# Digital Health Interventions to Improve Access to and Quality of Primary Health Care Services: A Scoping Review

**DOI:** 10.3390/ijerph20196854

**Published:** 2023-09-28

**Authors:** Daniel Erku, Resham Khatri, Aklilu Endalamaw, Eskinder Wolka, Frehiwot Nigatu, Anteneh Zewdie, Yibeltal Assefa

**Affiliations:** 1Centre for Applied Health Economics, School of Medicine, Griffith University, Nathan, QLD 4111, Australia; 2Menzies Health Institute Queensland, Griffith University, Nathan, QLD 4111, Australia; 3School of Public Health, The University of Queensland, Brisbane, QLD 4072, Australia; r.khatri@uq.edu.au (R.K.); a.sinshaw@uq.net.au (A.E.); y.alemu@uq.edu.au (Y.A.); 4College of Medicine and Health Sciences, Bahir Dar University, Bahir Dar, Ethiopia; 5International Institute for Primary Health Care in Ethiopia, Addis Ababa, Ethiopia; eskinder.wolka@iphce.org (E.W.); frehiwot.nigatu@iphce.org (F.N.); anteneh.zewdie@iphce.org (A.Z.)

**Keywords:** digital health, mHealth, eHealth, primary healthcare, universal health coverage

## Abstract

Global digital technology advances offer the potential to enhance primary health care (PHC) quality, reach, and efficiency, driving toward universal health coverage (UHC). This scoping review explored how digital health solutions aid PHC delivery and UHC realization by examining the context, mechanisms, and outcomes of eHealth interventions. A comprehensive literature search was conducted, capturing qualitative and quantitative studies, process evaluations, and systematic or scoping reviews. Our analysis of 65 articles revealed that a well-functioning digital ecosystem—featuring adaptable, interoperable digital tools, robust Information and Communications Technology foundations, and enabling environments—is pivotal for eHealth interventions’ success. Facilities with better digital literacy, motivated staff, and adequate funding demonstrated a higher adoption of eHealth technologies, leading to improved, coordinated service delivery and higher patient satisfaction. However, eHealth’s potential is often restricted by existing socio-cultural norms, geographical inequities in technology access, and digital literacy disparities. Our review underscores the importance of considering the digital ecosystem’s readiness, user behavior, broader health system requirements, and PHC capacity for adopting digital solutions while assessing digital health interventions’ impact.

## 1. Introduction

The 2030 Sustainable Development Goals emphasize having everyone receive the quality health services they need without any financial hardship [1]. Achieving universal health coverage (UHC) entails making significant progress on efficient, accessible, quality and equitable health services, while ensuring financial risk protection [2]. Primary health care (PHC), which first came to the fore with the 1978 Alma Ata declaration [3], provides the programmatic engine for UHC in most contexts and countries [4]. While the COVID-19 pandemic has resulted in global disruptions in the provision of essential health services including those provided at the PHC level [5], the public health measures put in place (e.g., lockdown, social distancing) have forced to shift the paradigm of service delivery model and provided an opportunity to accelerate the adoption and implementation of digital health solutions [6].

Digital health is an overarching term that comprises eHealth (e.g., telemonitoring, tele- and video- consultations, mHealth, electronic health records) and emerging technologies, including the use of computing sciences in the fields of artificial intelligence, big data, and genomics. The World Health Organization (WHO) defined electronic health (eHealth) as the use of information and communications technology (ICT) in support of health and health-related fields, including health care services, health surveillance, health literature, and health education, knowledge, and research [7].

The digital health infrastructure and its integration into PHC services vary greatly among countries, often influenced by the economic status, health priorities, and technological advancements of a region. Notably, the increasing ubiquity of mobile devices and internet connectivity offers a unique opportunity to leverage digital health solutions even in resource-constrained settings. Recent data indicate that more than 8 in 10 people in developing countries own a mobile phone, and nearly half the global population uses the Internet [7,8]. This widespread accessibility to digital platforms underscores the potential reach and impact of digital health interventions. Several countries have proactively responded to this rapidly evolving digital landscape. Approximately 87% of the World Health Organization member states have developed national policies or strategies geared toward eHealth, telemedicine, or the broader domain of digital health [7]. However, the degree of implementation and maturity of these strategies differ widely. While some countries boast advanced digital health ecosystems with comprehensive integration of electronic health records and telemedicine services, others are in the nascent stages, piloting innovative solutions tailored to their unique contexts. Factors such as internet penetration, mobile device accessibility, and data protection regulations play a pivotal role in shaping these digital health strategies. Recognizing the transformative potential of digital health, many international organizations and stakeholders are collaborating to bolster the digital health capacities of countries, especially in low- and middle-income regions.

The accelerated adoption of eHealth and mHealth platforms and the rapid change in the access to digital technologies have created a tremendous opportunity to expand the reach, quality, and efficiency of PHC service delivery and achieve UHC. Health systems can leverage advances in ICT to improve and maintain the continuity of service delivery post-COVID-19, such as by strengthening health management information systems and optimizing the functionality of shared electronic health records. Digital technologies also play an important role in advancing the core PHC tenet of a people-centered and integrated health service delivery model and community empowerment by improving information flows between patients and health workers, thereby shifting the nature of the patient–provider relationship. The WHO recently published a framework of e-Health for improved health service delivery [9], describing the potential contributions of e-Health to each of the health system attributes (i.e., service quality, efficiency, equity, accountability, sustainability, and resilience) at different levels—the individual, the service provider, the health-care organization, and the overall health system. In addition, newly emerging technologies, such as artificial intelligence and drones, have opened exciting possibilities and new avenues to improve the quality and accessibility of PHC services. However, without proper regulation and legislation, digital technologies may result in potentially harmful effects (e.g., mental illness in children associated with the digital revolution) and potential for worsening existing inequities [10].

In response to the 2018 World Health Assembly Resolution on Digital Health, the WHO conducted a review of the evidence for digital health interventions. While this review provided data regarding the impact of digital technologies on strengthening the overall health system, it was limited to a few selected interventions and did not provide insight on what works for whom and under what circumstances in the context of PHC [11]. This scoping review aimed to explore the underlying contexts and mechanisms in which digital health solutions contribute to improved PHC service delivery and the realization of UHC. We employed the WHO’s Framework of e-Health for improved health service delivery as an analytical framework to answer the following key questions: (i) Which eHealth solutions are adopted or implemented to increase access to PHC services and achieve UHC [8]? (ii) What is the role of contextual factors (i.e., why, how, for whom, and in what circumstances does it work?).

## 2. Materials and Methods

We carried out a comprehensive review of both published research and grey literature detailing digital health solutions related to PHC and UHC. This analysis was grounded in the PRISMA-ScR guidelines tailored for scoping reviews [12] (refer to Appendix A for details). A detailed overview of our approach is available in Erku et al. [13].

### 2.1. Data Sources and Search Strategy

Six online databases (PubMed, CINAHL, Web of Science, Cochrane Library, EMBASE, and Google Scholar) were explored, in addition to grey literature sources (PDQ-Evidence and mHealth Database), to find studies discussing the background, strategies, and results of eHealth strategies within PHC environments. Additionally, we undertook supplementary methods such as examining references and citations of the studies we found and using generic online searches to include potential articles missed in our initial database exploration. The search terms were formulated around three central themes: digital health, primary health care, and universal health coverage. These terms were adapted to fit the specific criteria of each database (for more details, refer to Appendix A). The use of Boolean operators and truncations was tailored based on the specific database. We considered articles written in English from the time the databases were created until December 2022 and we refreshed our search in August 2023. We did not set any restrictions regarding the publication period or the country of origin.

### 2.2. Eligibility Screening

We included editorials, opinion/position pieces, commentaries, process evaluations, qualitative and quantitative studies, program manuals, and systematic reviews that reported data on (i) the effectiveness eHealth solutions in improving PHC service delivery and UHC and (ii) contextual factors affecting the acceptability, feasibility, and implementation of eHealth solutions. We broadly categorized digital health solutions modalities into (i) technologies used to monitor, track, and inform health (e.g., mobile devices, such as smartphones and tablets, and clinical devices, mobile sensors, wearables, apps, social media), (ii) technologies used to enable health communication and provision of health services at a distance (e.g., client-to-provider and provider-to-provider telemedicine, targeted client communication), (iii) technologies used to collect, manage, and use health data (e.g., electronic medical records, electronic health records, artificial) [14]. We also included all system-level eHealth solutions targeting various levers of PHC and those modalities that have functions that cut across multiple health system attributes (e.g., health worker decision support tools, e-learning, stock notification and management tools, artificial intelligence). We considered all forms of delivery channels, including digital applications, SMS text messaging, voice calls, and interactive voice responses.

We excluded (i) eHealth solutions that were tailored for a specific health issue, (ii) conference notes or thesis overviews where the full text was not available. The identified articles were moved to COVIDENCE (by Veritas Health Innovation Ltd., Melbourne, Australia). Two reviewers separately checked all the titles, summaries, and the full content of the articles to ensure they fit our criteria. If the reviewers had different views, they discussed them until they agreed. Our search process is detailed in Figure 1. We used the Mixed Methods Appraisal Tool (MMAT) [15] to assess the quality of the studies we included, be they qualitative, quantitative, or a combination of both. This was to help us better understand the results based on the study’s quality, not to decide which studies to include.

### 2.3. Data Extraction and Synthesis

Information from each article was gathered and analyzed using both deductive and inductive methods. This process of data extraction was recurring, involving continuous discussions and agreements within the research team regarding the approach to data extraction and the initial analytical structure. We collected the study features into a table to provide a succinct summary of the digital health solutions that were included, focusing on their context, methods, and results. We recorded details about the study like the authors, publication year, objectives, design, and participant traits, as well as the main conclusions such as the goals, kinds, and scope of eHealth initiatives and their effects, including the ways eHealth affected PHC aspects and results related to UHC.

The extracted data were categorized into and analyzed as what caused an outcome, through which mechanism, and under which context [16]. Digital health interventions may work in one context but not in others, and as such, an outcome is measured as the context and mechanisms in which the program is implemented. In this review, the concept of “context” entails the relational, system-level, and dynamic features that shaped the mechanisms through which a digital health intervention worked and whether the country of the digital health implementation was a low-, middle-, or high-income country. Mechanisms are “underlying entities, processes, or structures which operate in particular contexts to generate outcomes of interest” [17]. Outcomes from implementing digital health solutions are measured as the impact on achieving UHC through improving PHC service delivery and accessibility. When research conducted in similar settings or contexts showed varying results, we combined and analyzed the evidence to pinpoint potential reasons for the differences. Additionally, we compared evidence when data on digital health solutions from one study provided insights into the outcomes mentioned in another study.

## 3. Results

After excluding duplicate entries and publications that did not fulfill the selection criteria, we selected 65 articles from 16 different countries. This collection included systematic and scoping reviews [18,19,20,21,22,23,24,25,26,27,28,29,30,31,32], secondary analyses of trial and quasi-experimental data [33,34,35], quantitative surveys [7,36,37,38,39,40,41], qualitative studies [14,42,43,44,45,46,47,48], mixed methods studies [49,50,51,52,53,54,55,56,57,58,59,60], program evaluations, case studies, and opinion pieces [61,62,63,64,65,66,67,68,69,70,71,72,73,74,75,76,77,78,79]. The included studies’ characteristics are summarized in the Appendix A.

All cross-sectional studies we reviewed scored either average (score ≥ 65%) or above (score ≥ 75%) based on the Mixed Methods Appraisal Tool. In the qualitative studies, the data were obtained through detailed interviews, focus group discussions, or a mix of the two. Each study explicitly stated its research objectives and clarified where its data came from, whether from participants or specific recruitment sites. However, many studies did not provide clear insights into the researchers’ roles or the specifics of the interview questions. In the next section, we outline how digital health impacts the core qualities of a high-functioning health system and we explore the contexts and strategies that bring about these outcomes.

### 3.1. eHealth Foundations for Improved PHC Service Delivery

Digital health solutions are potential tools for strengthening PHC and improving care delivery. However, they are only as good as the foundations, (pre)existing services, and governance systems that are put in place. There is a significant difference among and within countries in terms of eHealth adoption and the existing digital ecosystem. In countries with strong ICT foundations, such as Australia, Canada, the United Kingdom, and the United States, PHC practice transformed and responded rapidly to the COVID-19 pandemic by instituting telehealth, and electronic records enabled change [57,63].

A detailed scoping review by Ndayishimiye et al. (2023) [30] on the role of digital health tools in the COVID-19 response showed that these tools were frequently employed for diverse functions, with better adoption in high-income countries. These included facilitating virtual healthcare, offering clinical assistance, overseeing care quality, tracing and monitoring coronavirus transmission, and managing the inventory of medicines and vaccines [30]. However, disparities in internet access limited their adoption in LMICs and PHC settings located in rural and remote areas within high-income countries [63,75]. The adoption and success of eHealth in achieving UHC are also contingent on the presence and comprehensiveness of national eHealth strategies and policies with an overarching aim of achieving UHC via digitally enabled PHC service delivery. A recent global survey conducted by the WHO reported that more than half of the responding Member States have an eHealth strategy [7]. In Kerala, India, digital tools proliferated rapidly during the COVID-19 pandemic and helped meet diverse patient needs within PHC settings due to the foundation of their high-performing health system, existing eHealth strategy, and high levels of intersectoral collaboration, engagement with the private sector, and community volunteers [71]. Certain features of PHC practices, such as facilities with an affiliation to an academic institution and facilities that are located in an urban environment, can also increase the likelihood of eHealth adoption [54].

### 3.2. Digital Health and PHC Service Delivery

We found eleven articles reporting data on the impact of digital health solutions on improving the reach, quality, and/or efficiency of PHC service delivery and managing its demand. Studies examined several eHealth modalities (e.g., telemonitoring, tele-, video-, and email consultations) and a range of outcomes (e.g., health-seeking behavior, provider workload, service delivery, and accessibility). While none of the studies provided quantitative findings for interpreting causal relationships, positive intervention effects on PHC service delivery were the most consistently reported in the literature. The identified studies provided some level of evidence on some potential mechanisms through which eHealth affects the utilization and/or delivery of PHC services, including community outreach, PHC provider adherence to clinical guidelines, quality of care including patient–provider interaction, client trust, and uninterrupted access to essential drugs and equipment. Some of these mechanisms are intermediate (e.g., uninterrupted access to essential drugs due to digital stock management leads to improved client trust and demand).

Findings from the included studies suggest that telemonitoring data empowered patients to take a more proactive approach to their health care [37,45]. In India, a digital health -enabled task-shifting intervention (from doctors to frontline health workers) improved PHC service accessibility and helped meet the community’s diverse healthcare needs [33,64]. A few studies reported unfavorable findings on the role of e-consultations. Banks et al. [42] reported that an e-consultation system implemented in the UK primary care settings did not improve patient access or staff workload. Similarly, Casey et al. reported that although an online consultation system called *TeleDoc* had been rolled out successfully in England, its overall impact on shaping working practices and service delivery appeared insignificant [51]. Several local contextual factors influenced the effectiveness, including eHealth characteristics (perceived value, data storage and governance), actors’ roles and relationships (government champions, stakeholder networks), implementation processes (evidence, operationalization), and context such as interoperability [58]. Community members’ trust in frontline health workers as competent to undertake their new roles was another contextual factor for the uptake of the new services, and legitimacy provided by higher level providers (e.g., PHC doctors) enabled the development of trust [33,70]. One study conducted in the Netherlands examined the use of email consultation and found it to be extremely low; it also reported that its use by patients was largely dependent on its provision by General Practitioners (GPs) [38].

While telemedicine was widely adopted across PHC in many countries during the COVID-19 pandemic for delivering services such as maternal and newborn healthcare, it was not optimally supported by guidelines, training for health providers, adequate equipment, reimbursement for the cost of connectivity, and insurance payments for care provided remotely [26,37]. Studies revealed that technical problems were common across remote monitoring technologies, and infrastructure issues would need to be addressed for these technologies to have a sustainable impact on PHC service delivery [34,43]. In addition, digital illiteracy (of providers and patients), inability to perform in-person physical examinations, cost, lack of non-verbal feedback and client-provider relationship, language barriers, and client distrust were some of the contextual factors affecting the adoption and use of telemedicine [37]. Implementation (e.g., scaling up, provider workloads) and innovation issues (e.g., relative advantage or efficiency, complexity, compatibility with the pre-existing system) determine the widespread routine adoption of eHealth and the overall impact on service delivery [45].

Mobile-based digital technologies (mHealth) are widespread and routinely used by many PHC services, with providers often using their mobile phones to register clients, track their health, make decisions about care, and share clinical information with specialists and other healthcare providers. A Cochrane systematic review conducted by Gonçalves-Bradley et al. [18] demonstrated that mobile technologies might reduce the time between presentation and management of a health condition when PHC providers use them to consult with specialists, thereby providing efficient and integrated PHC service delivery. The mechanisms through which these outcomes are achieved include new opportunities created by mHealth (i) in how health workers communicate and coordinate with each other, patients, and management [80] (ii) in establishing, strengthening and/or maintaining engagement and relationships with clients and communities, (iii) in relation to the portability and work schedule flexibility of mobile devices and the possibility to access patients in rural and remote areas without having to travel, (iv) in relation to the ability to use treatment and screening algorithms that were loaded onto the mobile devices. For PHC providers working in rural and remote contexts, mHealth solutions provided an efficient service delivery model as it is perceived to save traveling time, increase speed, and enable the easy coordination of care delivery through robust provider-to-provider or provider-to-organization information sharing. In a cross-sectional study involving 1446 pregnant women in rural Madagascar from 2015 to 2019, the implementation of the Pregnancy and Newborn Diagnostic Assessment (PANDA) mHealth system led to pregnant women attending their first antenatal care (ANC) visit earlier in their pregnancy [41]. The duration of the ANC visits was significantly associated with several risk factors, including age, education level, and experience of domestic violence. The mHealth system demonstrated potential in standardizing ANC visits and boosting patient willingness for early and consistent ANC attendance [41]. In a similar study conducted in Tanzania’s Mufindi district, the mHealth system PANDA was assessed for its acceptability in antenatal care among pregnant women and healthcare workers. The results showed that those in the implementation group were significantly more satisfied with ANC visits than those in the control group [35]. The PANDA system, recognized for its user-friendly interface, effectively enhanced the quality of ANC, strengthened the relationship between healthcare workers and pregnant women, and addressed language and literacy barriers [35]. Contextual factors related to costs (e.g., recharging phones), the health worker (e.g., perceived usefulness), the technology (e.g., ease of use), the health system (e.g., availability of training, technical support, resource constraints, and extent of the integration with pre-existing electronic health systems—interoperability), and the infrastructure (e.g., access to the network and electricity) affected the extent of adoption of mHealth by healthcare providers in PHC settings [21].

To successfully implement mobile-based digital technologies in PHC, context-specific implementation strategies are essential [78]. MomConnect is a mobile health messaging service and helpdesk implemented in South Africa, providing pregnant women information via a short message service (SMS). The evaluation of its implementation revealed improvements in the quality of the services (e.g., decreased drug stockouts, behavioral changes in health workers) [50] and improved the health-seeking behavior [52]. High-level government buy-in and leadership, complex multistakeholder partnerships, formal integration with the public health system through facility-based registration and leveraging existing ICT technologies, long-term commitment, and earmarked funding for core functions were all fundamental to the successful scaling of and sustainable implementation of *MomConnect* in South Africa [49,68,81]. The adoption of vertical, single-condition digital health in PHC settings was reported to impede the sustainability of eHealth solutions and contribute to inefficiency as a result of the implementation of multiple digital technologies in overlapping geographies [58].

The reliable and uninterrupted availability of health commodities such as medicines is fundamental to delivering PHC services. Digital tracking technologies provide low-cost solutions to drug distribution and stockout challenges, including the real-time assessment, tracking, and reporting of essential commodities. A change in the availability of drugs and equipment in PHC facilities appeared as one of the mechanisms through which eHealth affects the quality of PHC services [20]. The utility of mHealth-based digital tracking technologies is contingent on several contextual factors, including the presence of strong and meaningful partnerships with local stakeholders and authorities (ICT companies and service providers), the availability of stock-level data at all levels of the health system, the availability of technical support, training and ongoing maintenance, and the presence of provider incentives (e.g., phone credit) [20]. The lack of consistent and standard outcome measures and sufficient data on mechanisms and contexts meant that we could not make a comparative analysis between studies and draw credible inferences regarding the role of mHealth in stock management.

#### 3.2.1. Electronic Decision-Support Tools

A recent Cochrane review concluded that the evidence on whether mobile-based clinical decision support tools make PHC providers better at following the recommended practice and their effects on patients’ and clients’ behavior are unclear [19]. The studies included in this Cochrane review did not take a systems lens or use a quality-of-care framework. In South Africa, *e-PC101*, an electronic clinical decision support tool, improved the quality and delivery of PHC in under-resourced health systems by streamlining the process and providing opportunity to examine clients systematically, comprehensively, and thoroughly [73]. However, the routine use of e-PC101 was challenged by the need to balance comprehensive clinical assessments with heavy clinic demands and provider workloads [73].

#### 3.2.2. Electronic Health Records

Electronic health records (EHRs)—a digital space to record, store, and share client’s health information—are another form of eHealth strategies widely implemented across health sectors to improve healthcare service delivery, reduce medical errors, and achieve healthcare cost saving. While we found limited evidence on the role of EHRs in PHC contexts, a few published studies reported that EHRs are more likely to be utilized if client information is easy to exchange between providers, both within and across PHC facilities and other institutions [54]. Factors facilitating the successful implementation of EHRs in PHC settings include improved ICT infrastructure, PHC provider’s motivation and incentives to use them, perceived threat to provider autonomy, confidentiality concerns, EHR design quality and functionality [28,29,60].

#### 3.2.3. Artificial Intelligence, Machine Learning

The recent revolution in Artificial Intelligence (AI) and Machine Learning (ML) presents a unique opportunity for PHC to deliver an effective, efficient, and equitable service [72]. While it was more difficult to assess the impact of AI-based applications on the continuity and coordination of PHC, there was a consensus among the studies that AI has the potential to improve managerial and clinical decisions and processes, although unsupervised machine learning is currently not robust enough to be widely adopted without rigorous checks in place [25,55,72].

### 3.3. The Role of eHealth in Improving Community Engagement

While innovative digital solutions are increasingly being used to improve community participation, the current evidence base regarding its impact is limited. Only two reviews indicated improved community involvement and external accountability [31,32]. One notable approach through which eHealth might bolster community engagement is by promoting involvement in online peer support programs. A comprehensive review on the role of eHealth technologies in fostering patient engagement and community participation by Barello et al. (2022) [31] found that digital solutions, while progressively leveraged, often overlook the emotional dimension of engagement. Many eHealth interventions predominantly focus on either the cognitive or the behavioral facets, leading to a fragmented understanding of the holistic patient experience. Furthermore, despite eHealth’s overarching goal of enhancing patient proactivity, there is a prevailing passive approach in its design. The study also highlighted that innovative eHealth solutions show promise in amplifying community participation, especially through online peer support programs, but evidence on their comprehensive impact remains limited. This underscores the nascent stage of the research field and the need for multi-dimensional assessments in future eHealth interventions [31].

### 3.4. Unintended Consequences of eHealth in PHC Settings

Several studies provide evidence on eHealth’s potential negative effects, or spillovers, in the PHC context. While eHealth resulted in increased PHC visits, patient satisfaction, and provision of health prevention education, it also seemed to increase the costs or decrease the usefulness for some parameters, e.g., leading to the loss of meaningful engagement with the clients [21] and to an increased number of drugs prescribed [22]. While PHC providers and administrators held positive views about remote monitoring technologies, several studies reported concerns regarding the perceived reduced quality of care from fewer face-to-face patient visits, overtreatment (e.g., over-prescription), and data incompatibility with PHC’s electronic health records [27]. Another study conducted in Swedish PHC settings reported that although PHC providers perceived working with digital consultation as flexible with a high grade of autonomy, flexibility, and reduced workload, concerns were raised regarding the loss of clinical competence when working exclusively with digital consultation [44]. In the U.K., the asynchronous nature of electronic consultation-based assessment meant that GPs could not probe for further information and make clinical decisions, deferring many e-consultations to face-to-face or telephone consultations. Furthermore, e-consultations did not substitute for, but were, rather, an ‘add on’ to, face-to-face consultations, further duplicating their workload [59]. The authors attributed this to the poor system integration, with the platform sitting outside the practice ITC and relying on staff manually importing e-consultation details into the electronic patient record in practice systems [42]. Confidentiality, privacy, and ethical issues were frequently discussed within and across the included articles, especially regarding individuals accessing data and the ability to send patient data to other facilities or institutions without a patient’s permission or knowledge [57,76,78].

#### The Digital Divide: eHealth and Equity

Over the last decade, the use of and access to mobile phones has increased in poor, remote rural areas and improved PHC access and delivery [48]. However, the differential interest in and/or the access to digital technology has important equity implications, with compounding inequalities between people living in regions that have ICT access and digitally excluded groups (those that do not have access or have restricted access). For example, in a study conducted by Dahlgren et al. [36] in Sweden, it was reported that the use of direct-to-consumer (DTC) telemedicine consultations was unevenly distributed across the population, with younger people with higher income, higher education, and born in Sweden being the most frequent users, raising some concerns about the equity implications [36]. The fact that some eHealth solutions (e.g., electronic consultations) [44] predominantly reach young, presumably healthy individuals raises equity issues for the elderly population with a higher burden of disease [44]. In the U.K., for people experiencing homelessness who may not have access to a phone, the move to remote telephone consultations highlighted the difficulties experienced in accessing PHC, as electronic consultations are often made remotely without taking into consideration both the clinical and the social factors underpinning health [46]. Furthermore, the rapid adoption and uneven expansion of telemedicine during the COVID-19 pandemic across different socio-economic groups further exacerbated the already existing inequalities in access to high-quality care [37]. The issue of multilingualism is of particular importance. For digital technologies to help deliver on the promises of an equity agenda that underpins UHC, linguistic diversity and cultural identity need to be ensured in the development and implementation of these technologies, so that health services and information are provided to people in the language they speak and in a culturally responsive way [7].

## 4. Discussion

Digital health solutions—including eHealth and the use of computing sciences in artificial intelligence and big data—are disruptive technologies that, in many ways, challenge the status quo of how healthcare services are provided. The overall evidence on the effectiveness of eHealth in the PHC context was mixed and inconclusive, which is consistent with other systematic reviews on digital health [20,23,27,80]. Our review highlighted several cross-cutting pathways and contextual factors that likely moderated the effects of digital health interventions in PHC, ranging from distal factors characterizing the organizational and wider health system attributes to proximal factors characterizing the individual, sociocultural, and facility-level attributes within which eHealth interventions are implemented.

In terms of distal factors, the wider health system attributes, including leadership and governance, regulatory and policy frameworks, and the level of decentralization of the health system (e.g., budget availability and autonomy within PHC facilities), were of key importance, as these factors determine the degree of financial and management autonomy at the PHC facility level. The proximal factors at the PHC facility level include individual-level factors (e.g., perceived benefit of eHealth, technical capacity, provider motivations and incentives) and the quality of facility infrastructure and existing PHC service delivery model before eHealth is adopted. These factors were important in shaping the provider’s response to eHealth solutions. In PHC facilities where skills, motivation, and digital literacy were higher and funding was adequate, the providers were more likely to adopt and apply eHealth technologies, with a subsequent impact on an efficient, quality, and coordinated service delivery and greater patient satisfaction. The presence of an optimal digital ecosystem including functioning and adaptable ICT infrastructure and conducive enabling environments is also crucial for eHealth interventions’ effective and efficient functioning. In countries where the current digital ecosystem and enabling environments are not mature enough to accommodate eHealth interventions, the adoption of eHealth interventions to facilitate PHC coordination and continuity is limited and is often fragmented. The proximal factors at the wider community level include pre-existing socio-cultural norms, geographical access to digital technologies, and digital literacy, which generally constrained the ability of eHealth interventions to improve demand. Other eHealth design features such as ease of use and interoperability (i.e., extent of integration with pre-existing electronic health systems) are also important contextual factors. Thus, when evaluating the impacts of eHealth on PHC service delivery, it is important to consider not only the technical functionality of the technology and the behavioral responses of the end-users, but also the wider health system’s needs and the digital ecosystem readiness including the capacity (individual, technological, organizational, and economic) available to accommodate the introduction of digital solutions [62]. As with any introduction of new and innovative approaches, eHealth requires providers and end-users to transition to embrace new practices. This can be achieved by conducting an all-inclusive stakeholder engagement in co-creating digital technologies, including design and implementation and improving educational and capacity-building efforts [7,78].

The rise in eHealth solutions over the past decade presents both opportunities and challenges for global public health. While the expansion of mobile phone usage in impoverished, remote regions has the potential to augment PHC access, concerns about the digital divide are evident [82]. Findings from our review indicate an inequitable distribution of eHealth services, where the more privileged demographics are the predominant users of such technologies. The COVID-19 pandemic further intensified these disparities, with the rapid, inconsistent growth of telemedicine highlighting pre-existing inequities in healthcare access among different socio-economic groups [66]. Additionally, the matter of linguistic inclusivity is pressing. For eHealth to truly uphold the equity tenets of UHC, it is essential to integrate linguistic and cultural considerations in digital health solutions. Yet, a mere third of nations with eHealth strategies in 2018 have incorporated multilingualism, pointing to a significant oversight in addressing the diverse needs of global populations [7]. To bridge the existing gaps, policymakers and stakeholders should prioritize the development of multilingual and culturally sensitive platforms, while facilitating the access to technology in economically disadvantaged and remote areas [82,83]. Moreover, integrating feedback mechanisms and regular equity audits can further ensure that eHealth solutions are evolving in a direction that caters to the diverse needs of the population, promoting a universally beneficial and cohesive healthcare environment [82].

The adoption of digital health interventions in PHC also brings forth other unique challenges, particularly concerning security and potential malpractice [84,85]. For instance, a local clinic using a mobile application to schedule appointments or monitor patient vital signs must ensure that the patients’ personal and health information remains confidential and secure. A breach in such environments could not only jeopardize patient trust but also deter them from seeking timely medical care. This emphasizes a need for systems that not only enhance PHC delivery but also embody the highest standards of data security. In addition to comprehensive training for PHC providers, continuous monitoring, updates, and quality assurance mechanisms must be embedded within these digital interventions in order to maximize the benefits of digital tools while minimizing the potential risks [85].

This scoping review assessed how and why digital health solutions implemented in PHC contexts result in intended or unintended outcomes by exploring the underlying mechanisms and contextual and program design moderators. Although we employed rigorous and standard approaches to describe and explain how and why digital health solutions work (or fail to work) to produce varied UHC-related outcomes in PHC settings, our review is not without limitations. Given that the majority of the included studies were scoping reviews, observational, and/or qualitative studies, we attempted to describe relationships between interventions (i.e., eHealth) and outcomes (service delivery, efficiency, quality, accessibility) rather than attributing any causal effects to eHealth and associated outcomes. Nearly all of the studies included in the review were not specifically designed to study pathways through which eHealth outcomes were achieved, and in many of these studies, it was often difficult to determine what factors moderated eHealth outcomes, mainly due to generalizability and design differences across eHealth modalities and the tendency of the included studies to conflate heterogeneous eHealth-specific factors (e.g., design feature) with non-intervention-specific moderators. Furthermore, most of the studies assessed eHealth intervention’s impact on one or two health system attributes, and only a few studies employed a systems lens to examine the overall impact of eHealth. Our review found no empirical data on the potential mechanisms or links between the use of digital and other attributes of PHC (e.g., governance and accountability, comprehensiveness), which might be partially attributed to our limited search strategy. Despite these limitations, the findings from this review can inform donors, policymakers, and implementers, helping them to design more effective digital technologies to strengthen PHC and achieve UHC.

## 5. Conclusions

Our review highlighted several cross-cutting pathways and contextual factors that likely moderated the effects of digital health interventions in PHC. When evaluating the role of digital solutions in PHC settings, it is important to consider not only the technical functionality of the technology and the behavioral responses of the end-users, but also the wider health system’s needs and the digital ecosystem readiness, including the capacity available at the PHC level to accommodate the introduction of digital solutions. There is a need for conceptual and/or methodological frameworks to better understand, classify, and examine the associative mechanisms of eHealth and PHC (or UHC-related) outcomes and understand how multifarious individual, organizational, technological, and system-level factors influence the performance of eHealth solutions.

## Figures and Tables

**Figure 1 ijerph-20-06854-f001:**
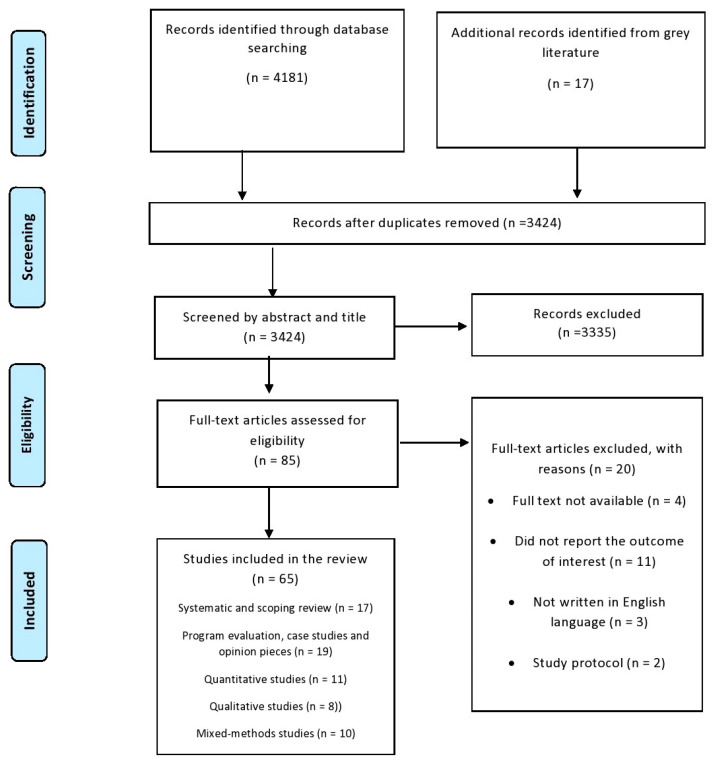
Preferred Reporting Items for Systematic Reviews and Meta-Analyses (PRISMA) flow diagram.

## Data Availability

All data supporting this study are provided in the manuscript and Appendix A.

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
