# Peer review of "Digital Health Interventions to Improve Access to and Quality of Primary Health Care Services: A Scoping Review"

_ijerph, 2023, doi:10.3390/ijerph20196854_

Round 1
Reviewer 1 Report
The conducted review addresses an interesting yet extremely broad and complex topic. However, the discussion is very limited and overlooks certain truly important and critical aspects related to digital health. In this sense, aspects concerning the security of these types of healthcare approaches are largely neglected, and especially the issue of malpractice due to incorrect usage.
Another element of great relevance is the issue of the use of digital health during the COVID pandemic, a aspect for which there are several examples in the literature, including the following.
Telemedicine as a medical examination tool during the Covid-19 emergency: The experience of the onco-haematology center of tor vergata hospital in Rome
Postorino, M., Treglia, M., Giammatteo, J., ...Cantonetti, M., Marsella, L.T.Author Response
- The conducted review addresses an interesting yet extremely broad and complex topic. However, the discussion is very limited and overlooks certain truly important and critical aspects related to digital health. In this sense, aspects concerning the security of these types of healthcare approaches are largely neglected, and especially the issue of malpractice due to incorrect usage.
Reply: Thank you for taking the time to review our manuscript and for drawing attention to the areas of concern. Our primary focus was to provide a comprehensive overview of the role and impact of digital health interventions in Primary Health Care (PHC). Given the vast nature of this domain, we had to make selective decisions on what specific issues to delve deeper into. Nevertheless, based on your feedback, we revisited this aspect and have now expanded our discussion. Specifically, we have integrated:
“The adoption of digital health interventions in PHC also brings forth another unique challenges, particularly concerning security and potential malpractice. For instance, a local clinic using a mobile application to schedule appointments or monitor patient vital signs must ensure that the patients' personal and health information remains confidential and secure. A breach in such environments could not only jeopardize patient trust but might also deter them from seeking timely medical care. This emphasizes a need for systems that not only enhance PHC delivery but also embody the highest standards of data security. In addition to comprehensive training for PHC providers, continuous monitoring, updates, and quality assurance mechanisms must be embedded within these digital interventions in order to maximize the benefits of digital tools while minimizing potential risks.”
- Another element of great relevance is the issue of the use of digital health during the COVID pandemic, a aspect for which there are several examples in the literature, including the following. Telemedicine as a medical examination tool during the Covid-19 emergency: The experience of the onco-haematology center of tor vergata hospital in Rome Postorino, M., Treglia, M., Giammatteo, J., ...Cantonetti, M., Marsella, L.T. International Journal of Environmental Research and Public Health
Reply: We appreciate the insightful comment and reference provided by the reviewer. Given that the suggested article was conducted in a hospital setting, and not within primary health care (PHC), we made the decision not to incorporate it into our analysis. Understandably, due to the vast scope of digital health, it was imperative for us to narrow our focus solely to PHC to maintain the specificity of our review. To address the broader perspective of digital health during the Covid-19 pandemic, we expanded our literature search and included a pivotal scoping review: “Ndayishimiye C, Lopes H, Middleton J. A systematic scoping review of digital health technologies during COVID-19: a new normal in primary health care delivery. Health Technol (Berl). 2023;13(2):273-284.”
“A detailed scoping review by Ndayishimiye et al. (2023) on the role of digital health tools in the Covid response showed that these tools were frequently employed for di-verse functions, with better adoption in high-income countries. These included facilitating virtual healthcare, offering clinical assistance, overseeing care quality, tracing, and monitoring coronavirus transmission, and managing the inventory of medicines and vaccines [30]. However, disparities in internet access limited adoption in LMICs and PHC settings located in rural and remote areas within high-income countries [59, 71].”
Reviewer 2 Report
The authors (AA) aim to explore how digital health solutions aid primary health care delivery and universal health coverage realization by examining the context, mechanisms, and outcomes of eHealth interventions.
This is an interesting scoping review useful to increase and to deepen our knowledge of the issue.
Addressing the following issues can make this manuscript eligible for publication.
Title
AA could drop “synthesis” in the title.
Introduction
The references are appropriate, but AA should add other references about their topic as reported below:
⁻ Benski AC, Schmidt NC, Viviano M, Stancanelli G, Soaroby A, Reich MR. Improving the Quality of Antenatal Care Using Mobile Health in Madagascar: Five-Year Cross-Sectional Study. JMIR Mhealth Uhealth. 2020 Jul 8;8(7):e18543. doi: 10.2196/18543.
⁻ Paduano S, Incerti F, Borsari L, Benski AC, Ernest A, Mwampagatwa I, Lilungulu A, Masoi T, Bargellini A, Stornelli F, Stancanelli G, Borella P, Rweyemamu MA. Use of a mHealth System to Improve Antenatal Care in Low and Lower-Middle Income Countries: Report on Patients and Healthcare Workers' Acceptability in Tanzania. Int J Environ Res Public Health. 2022 Nov 20;19(22):15342. doi: 10.3390/ijerph192215342.
- Borsari L, Stancanelli G, Guarenti L, Grandi T, Leotta S, Barcellini L, Borella P, Benski AC. An Innovative Mobile Health System to Improve and Standardize Antenatal Care Among Underserved Communities: A Feasibility Study in an Italian Hosting Center for Asylum Seekers. J Immigr Minor Health. 2018 Oct;20(5):1128-1136. doi: 10.1007/s10903-017-0669-2.
Why were these three references mentioned above not included in the scoping review?
Materials and Methods
Why did the authors decide to include systematic reviews instead of the original articles included in the systematic reviews?
Results
In some parts of the results section AA comment on the results as if they were under discussion. Review the discussion according to the following references:
Tricco AC, Lillie E, Zarin W, O'Brien KK, Colquhoun H, Levac D, Moher D, Peters MDJ, Horsley T, Weeks L, Hempel S, Akl EA, Chang C, McGowan J, Stewart L, Hartling L, Aldcroft A, Wilson MG, Garritty C, Lewin S, Godfrey CM, Macdonald MT, Langlois EV, Soares-Weiser K, Moriarty J, Clifford T, Tunçalp Ö, Straus SE. PRISMA Extension for Scoping Reviews (PRISMA-ScR): Checklist and Explanation. Ann Intern Med. 2018 Oct 2;169(7):467-473. doi: 10.7326/M18-0850. Epub 2018 Sep 4. PMID: 30178033.
McGowan J, Straus S, Moher D, Langlois EV, O'Brien KK, Horsley T, Aldcroft A, Zarin W, Garitty CM, Hempel S, Lillie E, Tunçalp Ӧ, Tricco AC. Reporting scoping reviews-PRISMA ScR extension. J Clin Epidemiol. 2020 Jul;123:177-179. doi: 10.1016/j.jclinepi.2020.03.016. Epub 2020 Mar 27. PMID: 32229248.
Line 233: Define acronym “GPs”
Author Response
The authors (AA) aim to explore how digital health solutions aid primary health care delivery and universal health coverage realization by examining the context, mechanisms, and outcomes of eHealth interventions. This is an interesting scoping review useful to increase and to deepen our knowledge of the issue. Addressing the following issues can make this manuscript eligible for publication.
- Title – AA could drop “synthesis” in the title.
Reply: “synthesis” is now removed from the title.
- Introduction – The references are appropriate, but AA should add other references about their topic as reported below: (1) Benski AC, Schmidt NC, Viviano M, Stancanelli G, Soaroby A, Reich MR. Improving the Quality of Antenatal Care Using Mobile Health in Madagascar: Five-Year Cross-Sectional Study. JMIR Mhealth Uhealth. 2020 Jul 8;8(7):e18543. doi: 10.2196/18543; (2) Paduano S, Incerti F, Borsari L, Benski AC, Ernest A, Mwampagatwa I, Lilungulu A, Masoi T, Bargellini A, Stornelli F, Stancanelli G, Borella P, Rweyemamu MA. Use of a mHealth System to Improve Antenatal Care in Low and Lower-Middle Income Countries: Report on Patients and Healthcare Workers' Acceptability in Tanzania. Int J Environ Res Public Health. 2022 Nov 20;19(22):15342. doi: 10.3390/ijerph192215342. (3)Borsari L, Stancanelli G, Guarenti L, Grandi T, Leotta S, Barcellini L, Borella P, Benski AC. An Innovative Mobile Health System to Improve and Standardize Antenatal Care Among Underserved Communities: A Feasibility Study in an Italian Hosting Center for Asylum Seekers. J Immigr Minor Health. 2018 Oct;20(5):1128-1136. doi: 10.1007/s10903-017-0669-2. Why were these three references mentioned above not included in the scoping review?
Reply: We appreciate the reviewer's insightful comments and the suggested articles. In light of your feedback, we conducted an updated literature search in August 2023, incorporating both new findings and some previously missed studies. We have incorporated two of the recommended articles on the PANDA mHealth pilot study conducted in Madagascar and Tanzania into our revised manuscript.
“In a cross-sectional study carried out with 1446 pregnant women in rural Madagascar from 2015 to 2019, the implementation of the Pregnancy and Newborn Diagnostic Assessment (PANDA) mHealth system was observed to encourage earlier attendance for the first antenatal care (ANC) visit. Notably, the duration of ANC visits was significantly correlated with specific risk factors like age, educational attainment, and prior experiences of domestic violence. The mHealth system showed its potential in not only standardizing ANC visits but also in enhancing patients' propensity for early and regular ANC attendance.”
“In a related study from Tanzania's Mufindi district, the acceptability of the PANDA mHealth system in antenatal care (ANC) was examined among pregnant women and healthcare providers. The findings indicated that participants in the intervention group expressed significantly higher satisfaction with their ANC visits compared to their counterparts in the control group. The PANDA system, with its intuitive design, not only elevated the overall quality of ANC but also fostered a stronger bond between healthcare professionals and pregnant women, while also bridging language and literacy gaps.”
Considering our stringent inclusion criteria, we have chosen not to incorporate the third article recommended.
- Materials and Methods – Why did the authors decide to include systematic reviews instead of the original articles included in the systematic reviews?
Reply: We appreciate the query regarding our methodological choices. Given the expansive scope of the topic, we anticipated that a direct search for original articles might yield an overwhelming number of results. Including systematic reviews allowed us to capture a synthesized and comprehensive overview of the available evidence on the subject. Additionally, systematic reviews often incorporate rigorous methodologies and broad search criteria, providing a holistic view of the current research landscape. However, to ensure comprehensiveness and reduce potential gaps, we also considered individual studies not encompassed within the included reviews when they were found relevant to our study objectives.
- Results – In some parts of the results section AA comment on the results as if they were under discussion. Review the discussion according to the following references: Tricco AC, Lillie E, Zarin W, O'Brien KK, Colquhoun H, Levac D, Moher D, Peters MDJ, Horsley T, Weeks L, Hempel S, Akl EA, Chang C, McGowan J, Stewart L, Hartling L, Aldcroft A, Wilson MG, Garritty C, Lewin S, Godfrey CM, Macdonald MT, Langlois EV, Soares-Weiser K, Moriarty J, Clifford T, Tunçalp Ö, Straus SE. PRISMA Extension for Scoping Reviews (PRISMA-ScR): Checklist and Explanation. Ann Intern Med. 2018 Oct 2;169(7):467-473. doi: 10.7326/M18-0850. Epub 2018 Sep 4. PMID: 30178033. And McGowan J, Straus S, Moher D, Langlois EV, O'Brien KK, Horsley T, Aldcroft A, Zarin W, Garitty CM, Hempel S, Lillie E, Tunçalp Ӧ, Tricco AC. Reporting scoping reviews-PRISMA ScR extension. J Clin Epidemiol. 2020 Jul;123:177-179. doi: 10.1016/j.jclinepi.2020.03.016. Epub 2020 Mar 27. PMID: 32229248.
Reply: We appreciate the reviewer's comment. Given the nature of our study, which employed a modified realist review approach, our analysis was intrinsically analytical. The context-mechanism-outcome configuration chosen for this scoping review necessitated a critical synthesis that not only discussed the results but also associated them with the mechanisms and outcomes of the specified digital health interventions. Thus, some aspects of our results section may resemble a discussion due to this analytical approach. We conducted our review adhering to the PRISMA Extension for Scoping Reviews (PRISMA-ScR) guidelines, which offers flexibility in synthesizing and presenting results in a manner that encompasses the depth and breadth of the data. We've referenced "Reporting scoping reviews-PRISMA ScR extension" to ensure rigorous methodology and reporting.
- Line 233: Define acronym “GPs”
Reply: “GPs” has now been defined as General Practitioners.
Reviewer 3 Report
A brief summary
This study examines the importance of digital technology in primary health care through a literature review, and presents digital health interventions and their outcomes, which are important implications for the global development of digital technology in health services.
Comment 1
The authors do not mention in the background of the study the digital infrastructure and policy guidance in different countries and regions, which leads the reader to believe that the role of digital technology is the same. I suggest that the authors add to the context the current digital infrastructure and national policies on the use of digital technologies in health services.
Comment 2
Line 329-334
The authors propose a role for e-health in increasing community engagement, but only one review was able to test the hypothesis. I suggest that the authors could add to the literature on the role of eHealth in improving community engagement or elaborate on the main points of the review.
Comment 3
In the discussion of the article, the issue of the "digital divide" should be discussed and strategies to address it should be proposed. I suggest that the authors elaborate on 3.5.1 in the discussion.
Author Response
Reviewer 3
A brief summary: This study examines the importance of digital technology in primary health care through a literature review, and presents digital health interventions and their outcomes, which are important implications for the global development of digital technology in health services.
- The authors do not mention in the background of the study the digital infrastructure and policy guidance in different countries and regions, which leads the reader to believe that the role of digital technology is the same. I suggest that the authors add to the context the current digital infrastructure and national policies on the use of digital technologies in health services.
Reply: We acknowledge the reviewer's valid concern. The digital health infrastructure and policy guidance indeed vary significantly from one country or region to another, reflecting the diverse challenges and priorities faced by different health systems. In response to this feedback, we have expanded our background section to provide a more comprehensive overview of the global digital health landscape:
“Digital health infrastructure and its integration into PHC services vary greatly among countries, often influenced by the economic status, health priorities, and technological advancements of a region. Notably, the increasing ubiquity of mobile devices and internet connectivity offers a unique opportunity to leverage digital health solutions even in resource-constrained settings. Recent data indicates that more than eight in 10 people in developing countries own a mobile phone, and nearly half the global population uses the Internet [7, 8]. This widespread accessibility to digital platforms underscores the potential reach and impact of digital health interventions. Several countries have proactively responded to this rapidly evolving digital landscape. Approximately 87% of World Health Organization member states have developed national policies or strategies geared toward eHealth, telemedicine, or the broader domain of digital health [7]. However, the degree of implementation and maturity of these strategies differ widely. While some countries boast advanced digital health ecosystems with comprehensive integration of electronic health records and telemedicine services, others are in the nascent stages, piloting innovative solutions tailored to their unique contexts. Factors such as internet penetration, mobile device accessibility, and data protection regulations play a pivotal role in shaping these digital health strategies. Recognizing the transformative potential of digital health, many international organizations and stakeholders are collaborating to bolster the digital health capacities of countries, especially in low and middle-income regions.”
- Line 329-334 – The authors propose a role for e-health in increasing community engagement, but only one review was able to test the hypothesis. I suggest that the authors could add to the literature on the role of eHealth in improving community engagement or elaborate on the main points of the review.
- Reply: Thank you for your feedback. We have incorporated additional information on the role of eHealth in improving community engagement as follows.
“While innovative digital solutions are increasingly being used to improve com-munity participation, the current evidence base regarding its impact are limited. Only two reviews indicated improved community involvement and external accountability [31, 32]. One notable approach through which eHealth might bolster community engagement is by promoting involvement in online peer support programs. A comprehensive review on the role of eHealth technologies in fostering patient engagement and community participation by Barello et al (2022) [31] found that digital solutions, while progressively leveraged, often overlook the emotional dimension of engagement. Many eHealth interventions predominantly focus on either the cognitive or behavioural facets, leading to a fragmented understanding of the holistic patient experience. Furthermore, despite eHealth's overarching goal of enhancing patient proactivity, there's a prevailing passive approach in its design. The study also highlighted that innovative eHealth solutions show promise in amplifying community participation, especially through online peer support programs, but evidence on their comprehensive impact remains limited. This underscores the nascent stage of the research field and the need for multi-dimensional assessments in future eHealth interventions [31].”
- In the discussion of the article, the issue of the "digital divide" should be discussed and strategies to address it should be proposed. I suggest that the authors elaborate on 3.5.1 in the discussion.
“The rise in eHealth solutions over the past decade presents both opportunities and challenges for global public health. While the expansion of mobile phone usage in impoverished, remote regions has the potential to augment PHC access, concerns about the digital divide are undeniable. Findings from our review indicate an inequitable distribution of eHealth services, where the more privileged demographics are the predominant users of such technologies. This uneven distribution poses equity challenges. The COVID-19 pandemic further intensified these disparities, with the rapid, inconsistent growth of telemedicine highlighting pre-existing inequities in healthcare access among different socio-economic groups. Additionally, the matter of linguistic inclusivity is pressing. For eHealth to truly uphold the equity tenets of UHC, it is essential to integrate linguistic and cultural considerations in digital health solutions. Yet, a mere third of nations with eHealth strategies in 2018 have incorporated multilingualism, pointing to a significant oversight in addressing the diverse needs of global populations. To bridge existing gaps, policymakers and stakeholders should prioritize the development of multilingual and culturally sensitive platforms, while facilitating access to technology in economically disadvantaged and remote areas. Moreover, integrating feedback mechanisms and regular equity audits can further ensure that eHealth solutions are evolving in a direction that caters to the diverse needs of the population, promoting a universally beneficial and cohesive healthcare environment.”
Round 2
Reviewer 2 Report
The authors (AA) have carefully addressed the reviewers' comments. Overall the changes made have improved the manuscript.